# Why are viral genomes so fragile? The bottleneck hypothesis

Nono S. C. Merleau[1], Sophie Pénisson[2,3], Philip J. Gerrish[4], Santiago F. Elena[5,6], Matteo Smerlak[1]*

1 Max Planck Institute for Mathematics in the Sciences, Leipzig, Germany, 2 Université Paris Est Créteil, CNRS, LAMA, Creteil, France, 3 Université Gustave Eiffel, LAMA, Marne-la-Vallée, France, 4 University of New Mexico, Albuquerque, New Mexico, United States of America, 5 Instituto de Biología Integrativa de Sistemas (I²SysBio), CSIC-Universitat de València, València, Spain, 6 Santa Fe Institute, Santa Fe, New Mexico, United States of America

* smerlak@mis.mpg.de

**Data Availability Statement:** Code for replication is available at https://github.com/strevol-mpi-mis/EvoEpi.

**Funding:** Funding for this work was provided by the Alexander von Humboldt Foundation in the

## Abstract

If they undergo new mutations at each replication cycle, why are RNA viral genomes so fragile, with most mutations being either strongly deleterious or lethal? Here we provide theoretical and numerical evidence for the hypothesis that genetic fragility is partly an evolutionary response to the multiple population bottlenecks experienced by viral populations at various stages of their life cycles. Modelling within-host viral populations as multi-type branching processes, we show that mutational fragility lowers the rate at which Muller's ratchet clicks and increases the survival probability through multiple bottlenecks. In the context of a susceptible-exposed-infectious-recovered epidemiological model, we find that the attack rate of fragile viral strains can exceed that of more robust strains, particularly at low infectivities and high mutation rates. Our findings highlight the importance of demographic events such as transmission bottlenecks in shaping the genetic architecture of viral pathogens.

## Author summary

Given that most mutations are deleterious, high mutation rates carry a significant evolutionary cost. To reduce this burden, an obvious evolutionary solution would be to reduce the fitness cost of mutations by becoming more robust; this solution is indeed selected in populations of constantly large size. Here, we show that when populations regularly experience bottlenecks, as viruses do upon transmission to a new host, a less obvious solution becomes more viable: namely, to *increase* the fitness cost of mutations so that unfit mutants are less likely to fix at each passage. This could explain why viruses—especially RNA viruses—do in fact have very fragile genomes.

## Introduction

From tobacco mosaic virus to poliovirus and SARS-CoV-2, some of the most consequential plant, animal and human pathogens are RNA viruses. In spite of their tiny genomes, these

framework of the Sofja Kovalevskaja Award endowed by the German Federal Ministry of Education and Research to M.S. Work in València was supported by Spain Agencia Estatal de Investigación - FEDER grant PID2019-103998GB-I00 and Generalitat Valenciana grant PROMETEO2019/012 to S.F.E. The funders had no role in study design, data collection and analysis, decision to publish, or preparation of the manuscript.

**Competing interests:** The authors have declared that no competing interests exist.

organisms find adaptive solutions to environmental challenges such as hosts' immune response, pervasive differences in susceptible cell types, switches in host and vector species, and antiviral drugs [1–3]. Such remarkable evolvability has been linked to the error-prone replication and short generation times of RNA viruses [4]. But high mutation rates are a double-edged sword: while replication errors provide the fuel necessary for rapid adaptation, they also increase the genetic load on viral populations, which in turn imposes a limit to genome size [5, 6]. Moreover, evidence gathered from diverse viral systems [7–11] shows that high mutation rates coupled with strong population bottlenecks (*e.g.* associated with airborne or fomite transmission events) turn on Muller's ratchet [12], resulting in the loss of fit genotypes [13, 14]. As fitness declines, populations risk experiencing a mutational meltdown, with low fitness genotypes unable to restore large population sizes and deleterious mutations accumulating at an ever increasing rate [15, 16]. How do RNA viruses manage to persist in spite of these challenges?

RNA viruses may have evolved specific mechanisms to maintain genome integrity in the face of high mutation rates [17–20]. Proposed mechanisms include complementation at high multiplicity of infection (MOI) during transmission *e.g.*, by physically aggregating viral particles [21, 22]; the use of stamping machine, rather than geometric, replication mechanisms [23]; the segmentation of viral genomes with reassortments during mixed infections or increased recombination rates, two simple forms of sex that reduce mutational load [24, 25]; or the co-opting of cellular chaperones (*e.g.*, heat-shock proteins) to assist at different stages of the replication cycle [26]. It is possible that all of these mechanisms (and others yet to be discovered) play a role in mitigating the damage done by mutations.

An extreme of such mitigation is one in which mutations cause no damage, i.e., they are neutral. And yet it is unrealistic to suppose that an organism can continue to accumulate mutations forever with no cost. This begs the question: how many mutations can an organism accumulate before a fitness cost is incurred? And how does that fitness cost then increase as further mutations accumulate? One common model of viral evolution, the neutral network model, supposes that a virus has a complex network of mutational neighbors (and neighbors of neighbors, etc) which incur zero fitness cost. Any mutant outside of this network, however, is non-viable. In this model, a viral population can mitigate the cost of mutation by evolving towards the center (away from the extremities) of such a network, where most mutations are neutral [27–30].

Counter-intuitively, the opposite strategy of *maximizing* mutational damage may also be a key component of the evolutionary response to low-fidelity replication. Both empirical and theoretical arguments support this hypothesis. First, viral RNA genomes are highly compacted, contain overlapping reading frames, encode for polyproteins that need to be precisely processed post-translationally, and express multi-functional proteins involved in different processes along the infection cycle; these structural properties all predict large deleterious effects of most mutations [5, 6], as is indeed observed [31]. Second, analyses of Muller's ratchet show that the risk of mutational meltdown is highest when deleterious effects are moderate [16]. This is because weak deleterious mutations negatively impact population fitness without being strongly selected against (Fig 1). This observation has led several authors to the conclusion that genetic fragility is in fact selected for in the high mutation rate regime of evolution [32–34].

In this paper we show that the population bottlenecks experienced by RNA viruses contribute to promoting fragile genetic architectures. Our contribution is twofold. First, we model population bottlenecks (and the genetic drift they induce, including Muller's ratchet) explicitly using multi-type branching processes [35, 36]; such bottlenecks are not easily modeled within more common approaches based on weak-mutation strong-selection limit [33] or quasi-species theory [32, 34]. Using general results in branching process theory, we derive expressions for the survival probability of a population through repeated bottlenecks as a function of the

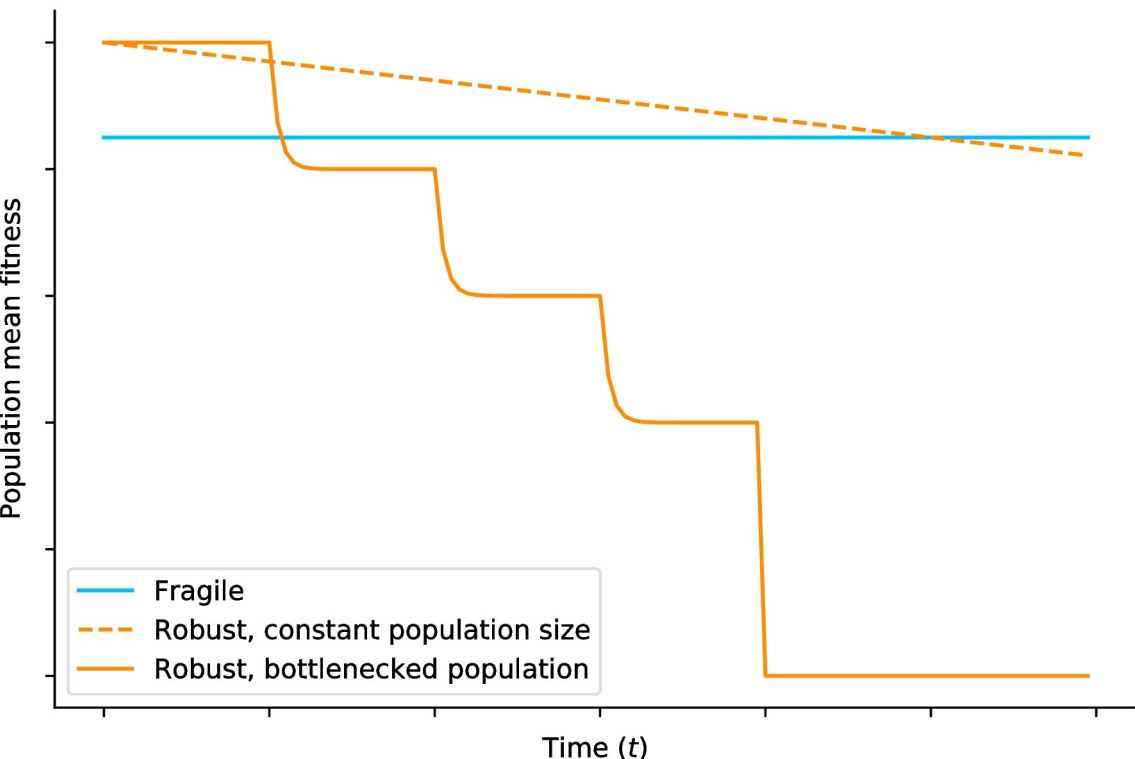

**Fig 1. Schematic of fitness dynamics.** Robust genomes (orange) initially have a lower mutational load, and therefore higher population mean fitness, than fragile genomes (blue). But they also fix deleterious mutations more frequently, leading to a gradual decline in population fitness (dashed line); this effect is more pronounced when bottlenecks weaken the strength of selection and accelerate the clicking of Muller's ratchet (continuous line).

deleterious effect of mutations, confirming that fragility can be advantageous in the long run. Second, we consider the epidemiology of genetic fragility using a simple agent-based compartmental model. Evolutionary epidemiology [37] is an emerging field focusing on the interactions between evolutionary and epidemiological dynamics that seeks to explain the evolution of virulence and other properties of pathogens. Here we ask under what epidemiological conditions a fragile strain can have a higher attack rate than a robust one. We find that two parameters determine whether or not this is possible: the mutation rate $u$ and the infection rate $\beta$ (both relative to the recovery rate), with high $u$ and low $\beta$ both favouring fragile viruses.

Our model builds upon the standard model of Muller's ratchet [13], under which all mutations have the same deleterious effect (in particular, none is lethal), and do not interact epistatically. However, positive epistasis among deleterious mutations has in fact been shown to be a pervasive phenomenon in compacted RNA genomes [38]. Likewise, the fraction of mutations that are lethal is usually large for RNA viruses [39]. We rationalize these simplifications by noting that relaxing them would further increase the long-term advantage of fragile genomes, reinforcing the argument for the bottleneck hypothesis.

## Results

### Muller's ratchet in expanding viral populations

Consider a small viral population with absolute fitness $w_0 > 1$ and genomic mutation rate $u$. Assume that all mutations are deleterious with the same effect $s_d$, such that an individual

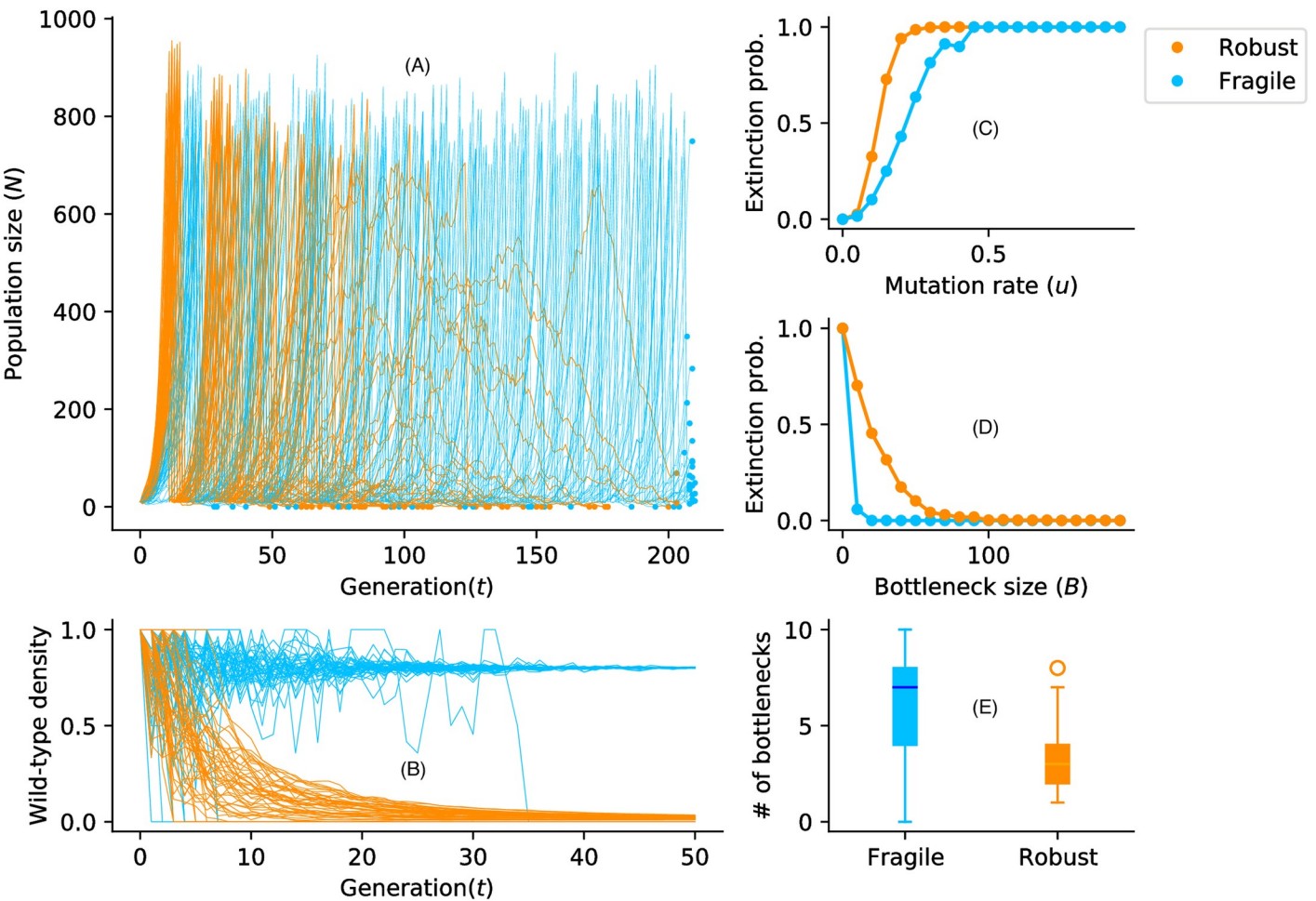

**Fig 2. Viral populations through multiple bottlenecks.** For an initial population size $N$ = 10 with fitness $w_0$ = 1.5 and a mutation rate $u$ = 0.2, viral particles reproduce, mutate, and die out. Once the populations reach the carrying capacity $C$ = 800, a sub-population of size $B$ = 10 is sampled and the branching process is restarted. (A) Populations of robust viruses grow faster but go extinct more often after multiple bottlenecks. (B) The wild-type (individual with fitness $w_0$) density in the populations over time. (C) Effect of mutation rate on the extinction probability for a fixed bottleneck size $B$ = 10. (D) Effect of $B$ for a fixed mutation rate ($u$ = 0.2) on the extinction probability. (E) Number of bottlenecks before extinction.

carrying $i$ mutations—an "$i$-mutant"—has fitness $w_i \equiv w_0(1 - s_d)^i$. The dynamics of such a population falls under two broad alternatives: either it goes extinct through demographic fluctuations, or it grows to unbounded sizes with an asymptotic mean fitness $1 < \bar{w}_\infty \leq w_0$; the latter outcome is only possible if $w_0 e^{-u} > 1$. This is illustrated in Fig 2 for a low value of $s_d$ (a "robust" type) and a high value of $s_d$ (a "fragile" type).

Key to the fate of the population is the onset of Muller's ratchet, *i.e.* the extinction of fit genotypes through genetic drift in small populations. The ratchet mostly clicks during the early states of the expansion, when the population size is smallest and extinction is likely. But it may also start clicking later through some rare fluctuation; if this click is followed by another click, and then another, the population can start shrinking again into mutational meltdown. If the ratchet clicks too many times, namely more than $K = \max\{k: w_k e^{-u} > 1\}$ times, extinction is unavoidable: any mutant carrying more than $K$ mutations has absolute fitness smaller than one, and therefore generates a subcritical branching process.

Robust and fragile populations experience Muller's ratchet differently. For a robust genotype, mutations have a small deleterious effect, and are therefore under weak negative

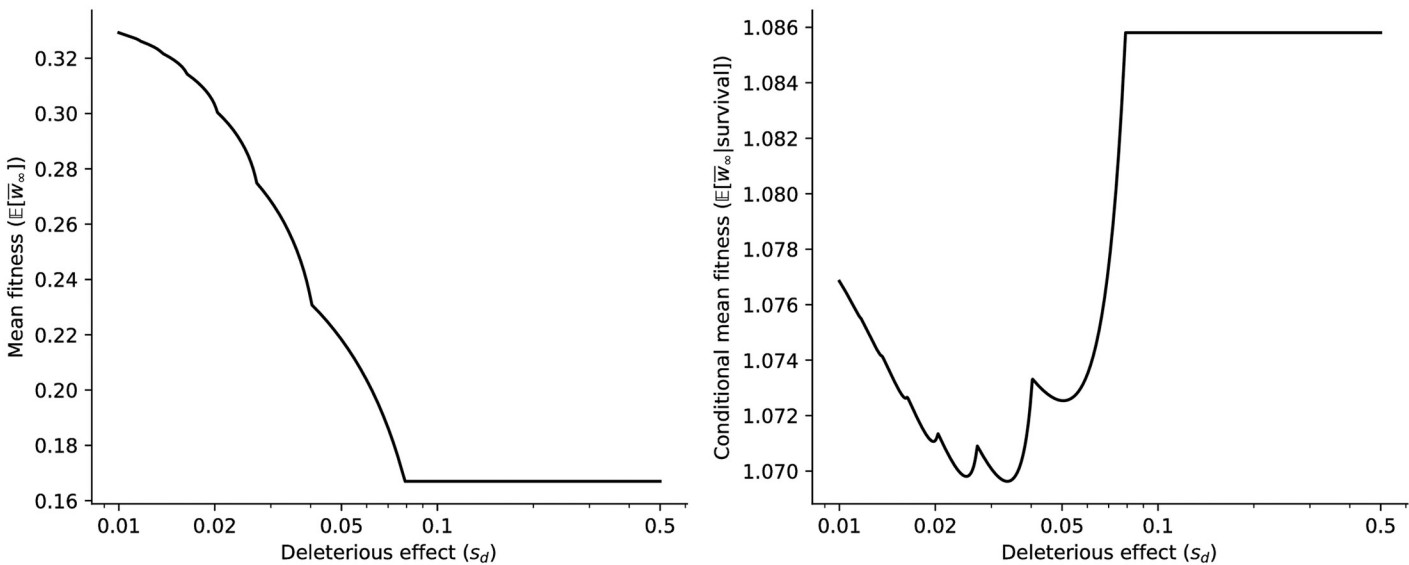

**Fig 3. Analytical results for expected asymptotic population mean fitness.** Both are unconditional (left) and conditional on survival (right) as a function of the deleterious effect of mutations $s_d$, for a population with $w_0 = 1.2$ and $u = 0.1$. While a more robust strain has a lower mutational load and therefore a higher population mean fitness (left), it is also more vulnerable to Muller's ratchet, implying that surviving populations tend to have lower mean fitness (right).

selection. This makes a click of the ratchet quite likely. On the other hand, each click comes at a low fitness cost, and so the population can withstand a relatively large number of clicks. Fragile populations, by contrast, are under strong negative selection against deleterious mutations, hence clicks are rarer; when they do occur, though, extinction becomes almost certain. Which fares better?

Using branching process theory we compute analytically the probability $p_k$ that the fittest surviving individual carries exactly $k$ mutations, *i.e.* that Muller's ratchet click $k$ times (Eq (1) in Methods). From this, we find that both the survival probability $p_{\text{surv}} = \sum_{k=0}^{K} p_k$ and the expected asymptotic population mean fitness $E(\bar{w}_\infty)$ (Eq (2)) are decreasing functions of $s_d$, consistent with the idea that, when all mutations are deleterious, mutational robustness is an evolutionary advantage. But there is a caveat: because Muller's ratchet clicks less frequently when genomes are fragile, the populations that do emerge from the expansion tend to be mutations-free. As a result, the asymptotic mean population fitness *conditional on survival* $E(\bar{w}_\infty \mid \text{survival})$ turns out to be non-monotonic in $s_d$, and in fact to be maximized for large deleterious effects (Fig 3).

## Survival of viral population through multiple bottlenecks

The adaptive value of virus' mutational fragility becomes apparent when we consider a succession of bottleneck-expansion cycles (Fig 2), corresponding to viral transmission followed by within-host replication. This process can be modeled as a Markov chain on the space of post-bottleneck populations. Let $B$ denote the size of the population after the transmission bottleneck. (In some cases, $B$ can be a small as 1 [40, 41]).

We model virus transmission as sampling without replacement from a surviving population as described above, resulting in a new founding population with composition $\mathbf{n} = (n_0, \cdots, n_K)$, where $n_i$ is the number of $i$-mutants and we omit the subcritical mutants whose lineage will go extinct with probability one (hence $\sum_{k=0}^{K} n_k \leq B$). After within-host expansion and transmission sampling, this population will give rise to a new founding population with composition

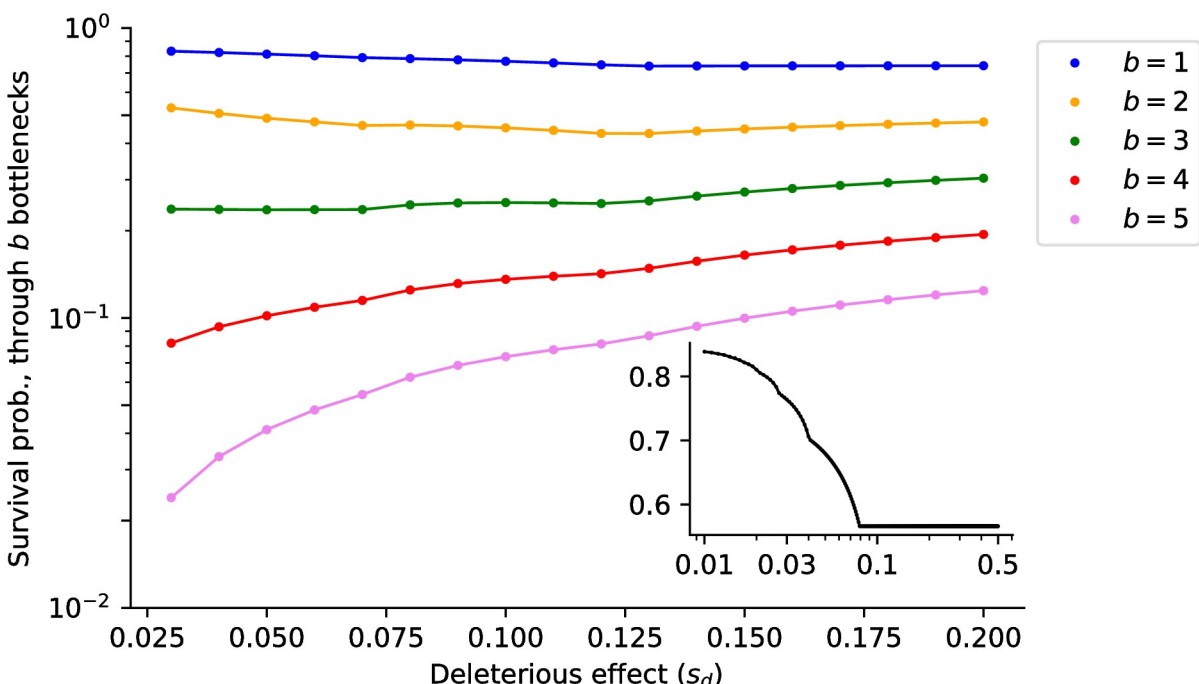

**Fig 4. Analytical results for the survival probability *vs.* mutational fragility.** For a population with $w_0 = 1.2$ and $u = 0.05$ undergoing $b$ bottlenecks of size $B = 5$, it is computed using the bottleneck-to-bottleneck Markov chain $P(\mathbf{n} \to \mathbf{m})$. The higher the number of bottlenecks a populations has to survive, the more fragility is selected for. The inset figure shows the survival probability for $b = 0$, $w_0 = 1.2$ and $u = 0.05$.

$\mathbf{m} = (m_0, \cdots, m_K)$ with a probability $P(\mathbf{n} \to \mathbf{m})$ given explicitly in (Eq 10 in S1 Text). From this Markov chain, we can compute the probability that a population can survive any given number of bottlenecks (Eq (4)).

Fig 4 shows the survival probability after up to five bottlenecks of size $B = 5$ as a function of the deleterious effect $s_d$. Although non-monotonic, this probability is maximized at large $s_d$ when the number of bottlenecks increases, *i.e.* extinction becomes less likely for more fragile genomes. The results in the previous paragraph explain why: fragile populations are protected against Muller's ratchet, hence each new infection starts from fit founders. By contrast, robust genomes accumulate deleterious mutations; after several transmission bottlenecks, the founder particles tend to have low fitness and become increasingly unlikely to give rise to surviving lineages. It is as if robustness promoted mutational meltdown on a longer time scale—a meta-population meltdown.

## Epidemiology of fragility

How would these effects play out in an epidemic outbreak? Would the lower propensity of fragile genomes to suffer meltdown make up for their lower initial population fitness? To investigate this question we consider a Susceptible-Exposed-Infectious-Recovered (SEIR) model defined as follows. When a susceptible $S$ meets an infectious individual $I$, a sample of the latter's viral population with size $B$ is transmitted and $S$ becomes exposed $E$. After this event, the within-host viral population carried by $E$ can either (*i*) go extinct, in which case $E$ returns to the susceptible compartment ($E \to S$) or (*ii*) grow exponentially during an incubation period $\tau$ until it reaches a critical threshold $C$ which makes the host infectious ($E \to I$). Which is more likely depends on the viral genetic parameters ($w_0$, $s_d$, $u$) and, as shown in the previous section, on the numbers of supercritical viral particles transmitted to the new host,

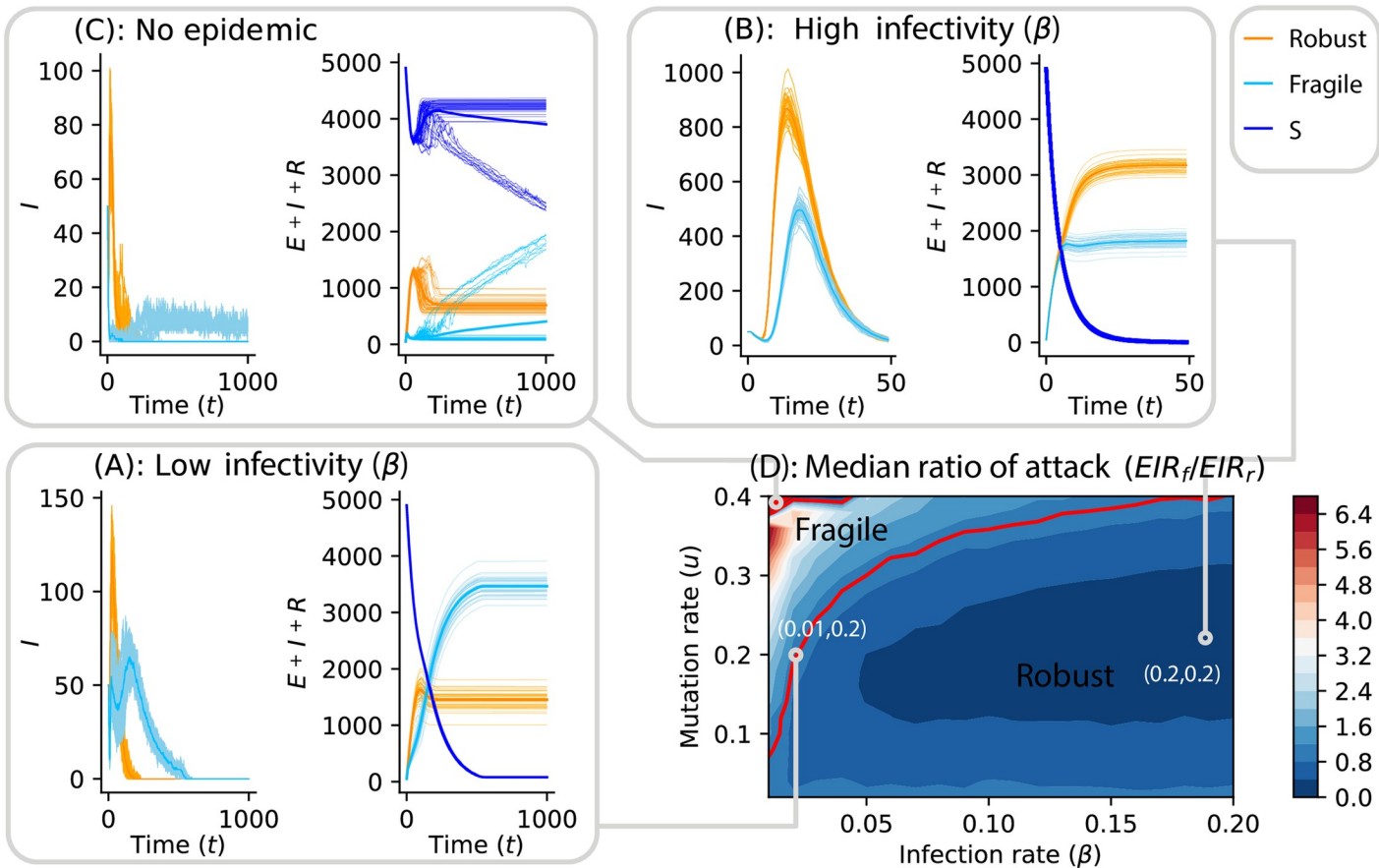

**Fig 5. Agent-based simulation of the SEIR model.** (A) At low infection rate ($\beta = 0.01$) and high mutation rate $u = 0.2$, the robust genomes are more infectious in the beginning of the epidemic than the fragile ones, but as time goes on, it turns out that the fragile genomes become more virulent. (B) Using the same mutation rate and the highest infectivity gives the robust genome a chance to become the most infectious because it reproduces faster, and its population undergoes fewer bottlenecks. (C) At low infection rate and very high mutation rate ($u = 0.4$), the disease does not spread in the population, due to the high extinction probability of the viral populations. (D) Median ratio of attack rate (which is the total number of fragile infections divided by the robust ones) across 50 runs for each ($\beta$, $u$) parameters. The red line shows the region of parameter ($\beta$, $u$) where both fragile and robust strains have equal attack rate.

described by some vector **n**. The incubation time is random as well, with a probability distribution depending on **n** and on the number of clicks of Muller's ratchet during the expansion, see Eq (5). (We show in S1 Text that this distribution can be approximated by a Gamma distribution depending on these parameters.) The host population is assumed to be well-mixed (no spatial structure): at each time step and for each pair ($S$, $I$), transmission occurs with a probability $\beta$. Recovery in turn takes place at a rate $\gamma$.

Fig 5 presents the results of simulations where a susceptible population of size $N = 5000$ is seeded with 100 infectious individuals, half of which carry a robust viral strain ($s_d = 0.05$) and the other half a fragile one ($s_d = 0.9$). Three regimes emerge depending on the transmission rate $\beta$ and the mutation rate $u$ (at fixed bottleneck size $B = 5$). When infectivity is high and the population quickly becomes completely infected, the robust strain fares better due to its higher initial fitness and faster growth rate. When infectivity is low, however, fragile strains prove to have a much higher attack rate (We define the attack rate of a strain as the number of hosts infected by that strain.). As noted in Fig 5A, the robust strains are more virulent in the early stages of the epidemic. But as time goes on, the robust genomes loose fitness and increase their post-bottleneck extinction probabilities. That is also the reason why there is a small decrease in

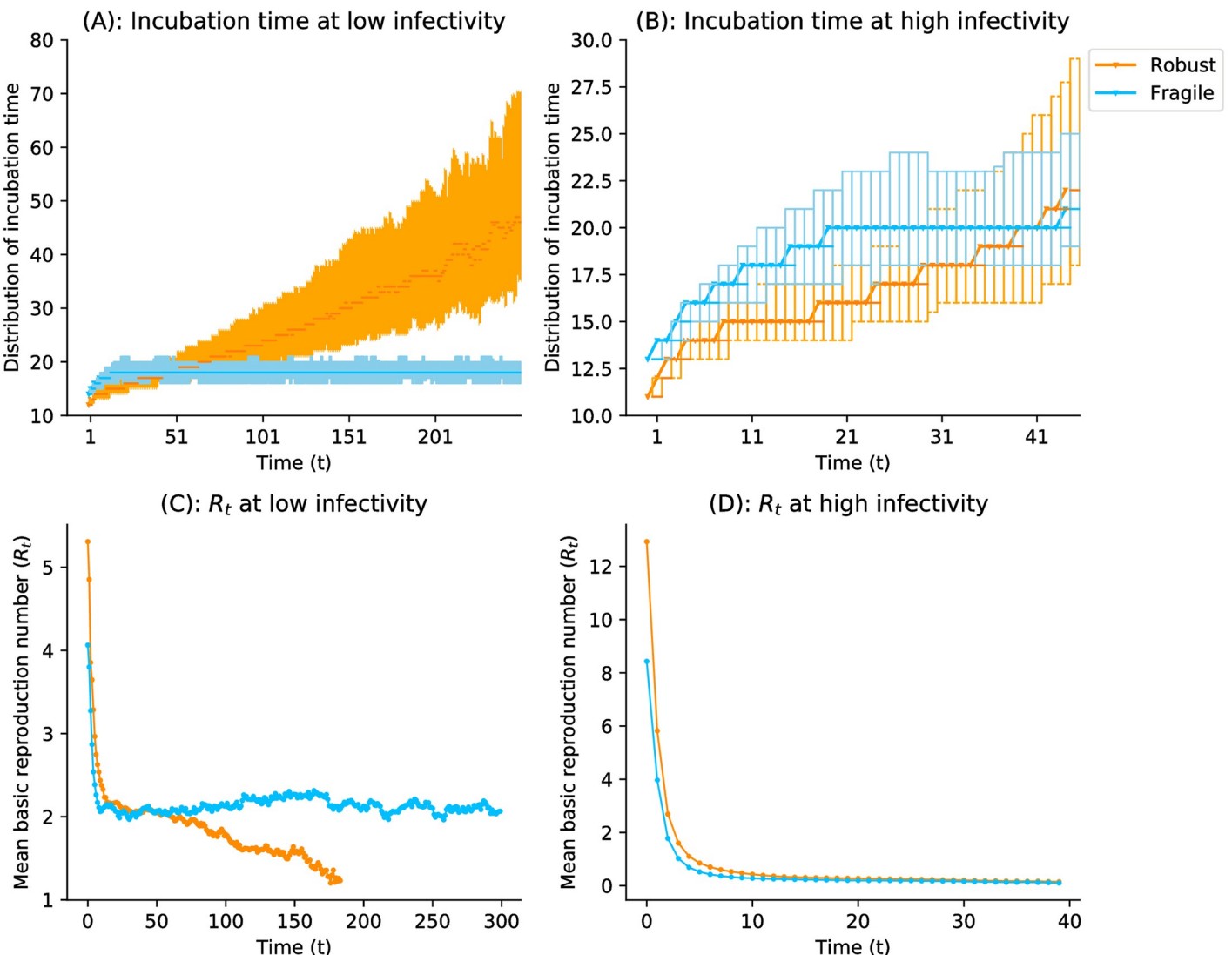

**Fig 6. Incubation time distributions and mean basic reproduction number at low ($\beta$ = 0.01) and high ($\beta$ = 0.2) infectivity, with viral parameters $w_0$ = 1.5, $u$ = 0.1 and $B$ = 5.** (A) The epidemic lasts longer and the robust viral populations lose fitness. The later transmitted viral particles take a longer time to grow up to $C$. In contrast, the incubation times of the fragile strains stay on average constant. (B) On short-time epidemic, the mean incubation time of the robust strain stays smaller than the fragile ones. (C) The epidemic lasts longer, wherein the beginning, the reproduction number of the robust strain is higher than the fragile one but later on, when the robust strain's $R_t$ decreases, the fragile one stays constant. (D) The virus takes over the population in the early stage of the epidemic when the robust strains are the most virulent.

the total number of robust genome infections in ($E + R + I$)-populations, as exposed hosts return to the susceptible compartment (Fig 5A). At a late stage ($t > 200$), mutationally robust viruses are too degraded to make their host infectious. Finally, very high mutation rates and low infectivities do not allow for the epidemic to spread, as all viral populations (robust and fragile) quickly go extinct. These findings are summarized in Fig 5D in terms of the median ratio of attack rates across 50 runs for each parameter set.

Underlying this contrast is the distribution of incubation times, which differs for the robust and fragile strains (Fig 6A and 6B). The same observation can be drawn by looking at the mean basic reproduction number of both robust and fragile strains at low infectivity (and

respectively at high infectivity) (see Fig 6C and 6D). Over time, the basic reproduction number $R_t$ of fragile strains stays constant while the robust one decreases.

## Discussion

There is a growing appreciation for the fact that population bottlenecks are not just a fundamental aspect of the life cycle of viruses, but that they also play a key role in their evolution [42, 43]. The effect of bottlenecks is not always detrimental: bottlenecks can effectively remove cheaters, *e.g.* defective interfering viruses [44], or enhance the effectiveness of selection if beneficial alleles act in *trans* [45]. In addition, genetic bottlenecks can facilitate traveling across the characteristically rugged fitness landscapes of RNA viruses [38], where it is easy for viruses to become trapped at suboptimal fitness peaks [46, 47]; by relaxing the intensity of selection, bottlenecks enable the exploration of new regions of the landscape.

In this paper we have explored another aspect of highly mutable populations subjected to periodic bottlenecks: they experience a strong evolutionary pressure towards genetic fragility. Earlier work has established that intermediate deleterious effects $s_d$ maximize the strength and speed of Muller's ratchet [16, 32]; similar results have been reported more recently in terms of "ratchet robustness" [48] or "drift robustness" [33]. We find this U-shaped pattern in the context of expanding populations as well, *e.g.* for the asymptotic population mean fitness given survival (Fig 3); the unconditional population mean fitness, by contrast, always decreases with fragility, which is consistent with another analysis of the advantage of mutational robustness [34]. By modelling changes in population sizes with branching processes, we allowed a more complete picture to emerge. In this picture, the evolution of high neutrality [27, 49] is not incompatible with selection for maximally deleterious mutations [32] and anti-redundancy [34]. An agent-based SEIR epidemiological model further reveals the determinants of genetic fragility, highlighting the importance of epidemic transmission parameters.

The evolution of viruses is often described in terms of fitness landscapes and their topographies. Our findings highlight the limitation of this picture: the motion of evolving population in genotype space depends on the structure of the genotype-phenotype-fitness mapping, but also on mutation rates and life cycle parameters such as the frequency and stringency of population bottlenecks. As a result, the evolution of mutational robustness—or mutational fragility—cannot be construed solely as the search for an optimal region in the fitness landscape, be it the highest peak or the flattest plateau. Our multi-type branching process approach is suggestive that a composite definition of fitness might be more predictive of evolutionary success in the present context, namely, a definition that takes account of both offspring number (Malthusian fitness) and long-term survival probability. The augmented evolutionary relevance of this definition to the present context is manifest in the comparison between the left and right panels of Fig 3.

To be sure, our model relies on simplifying assumptions, mainly pertaining to the nature of underlying mutational landscapes and to epidemiological details. The assumption that all mutations having the same deleterious effect is a common one that simplifies the mathematics. Fitness effects of deleterious mutations are more realistically modeled as a random variable with a continuous, heavy-tailed distribution (*e.g.* the Gamma and Weibull distributions have been previously used to satisfactorily fit experimental data [50, 51]) or a U-shaped distribution to incorporate lethal mutations. As already noted, we make another common assumption that does not necessarily hold for real viral genomes, namely, the independence of mutational effects.

Evidence pervasively suggests that positive epistasis is the norm for compacted viral RNA genomes [38], including many instances of compensatory mutations. Indeed, it was shown long ago that if deleterious alleles interact synergistically, they are more efficiently removed from the population and thereby slow the advance of Muller's ratchet [52]. Finally, our evolutionary epidemiological approach ignores co- or super-infections, wherein multiple viral strains infect the same host. This last point is subtle. On the one hand, multiple infections allow competition among strains to occur within hosts as well as across hosts, which may reinforce the advantage of fragile strains in the low $\beta$ regime. On the other hand, multiple infections allow the sharing of gene products among different genotypes within a cell, thus compensating for deleterious effects and eventually contributing to the accumulation of more mildly deleterious mutations in the population [53].

If position on the robustness-fragility spectrum is a heritable and variable trait, then that trait is subject to adjustment by indirect selection [54]. How this trait evolves will be determined by a tradeoff between the individual vs population cost of mutation. Under low mutation rates, selection on this trait at the individual and population levels can coincide. Under higher mutation rates and low effective population sizes (as studied here), however, they can diverge, favoring increased but unstainable robustness at the individual level, but favoring increased and sustainable fragility at the population level.

This tradeoff, however, becomes more complex in changing environments: if a change in environment is lethal to the fragile genome, and if "evolutionary rescue" [55–57]—through, for example, the viability of an immediate mutational neighbor—is not feasible, this may tip the balance toward robustness. This interplay between environmental change and the evolution of genetic architecture has been dubbed plastogenetic congruence [58] in the context of RNA viruses which can face dramatic, unpredictable and fast fluctuations in their environments. This interplay between environment and genetic architecture has been observed in several experimental studies [59, 60].

A study of particular relevance to our work [61] found that poliovirus at high multiplicity of infection (MOI) contained more genetic variability, including many neutral mutations, and were more mutationally robust. By contrast, populations at low MOI evolved greater fragility. These findings are entirely consistent with our theoretical findings here.

This study, however, introduced a further complexity that added some realism: the poliovirus populations were then used to infect hosts that differed in their susceptibility to infection (a heterogeneous environment). Intriguingly, and counter to arguments put forth by [58], it was the more fragile viruses that were more successful adapting to the more restrictive novel hosts. This finding supports the notion that evolutionary rescue of fragiles may be effective enough in RNA viruses that fragiles can even be superior in heterogenous or changing environments.

Clearly, there is more work to be done to determine how environmental heterogeneity in space and time tip the balance in favor or robustness vs fragility. And incorporating beneficial mutations into our analyses would add further realism. Beneficial mutations are typically rare and, when mutation rates are high, can arise on genetic backgrounds contaminated by deleterious mutations (a "ruby in the rubbish" [62]). It would be interesting to determine which has the higher probability of fixation: 1) a beneficial arising on a robust, only slightly eroded, background, or 2) a beneficial arising on a fragile background, where it arises on equal footing with the rest of the population but at an effectively reduced rate. These considerations warrant a follow-up study.

Overall, our findings can be summarized with a classic metaphor. Robust genomes are hares: they grow fast but accumulate mutational damage through Muller's ratchet, which jeopardizes their potential for long-term survival. Fragile genomes are turtles: they grow

more slowly but weather bottlenecks more reliably and have higher long-term survival probabilities. Turtles may seem weak individually, but as a group they have survived for hundreds of millions of years. Similarly, fragile viral genomes are individually vulnerable to deleterious mutations; at the meta-population level, however, they may hold the key to evolutionary resilience.

## Materials and methods

The evolution of viral populations experiencing Muller's ratchet and going through bottlenecks is modeled by a multitype branching process, in which a "type" corresponds to the number of accumulated deleterious mutations. We refer to S1 Text for a detailed description of the model and proofs of the following results.

Starting with $\mathbf{n}$ supercritical mutants, the extinction probability $p_{\text{ext},k}(\mathbf{n})$ of all types up to type $k$ in the population is given by the element-wise product $\mathbf{q}_k^{\mathbf{n}}$, where $\mathbf{q}_k$ is the smallest solution to a fixed-point equation involving the generating function of the branching process. Muller's ratchet click probability $p_k(\mathbf{n})$ that the fittest surviving individuals carry $k$ mutations is then

$$p_k(\mathbf{n}) = p_{\text{ext},k-1}(\mathbf{n}) - p_{\text{ext},k}(\mathbf{n}) \qquad (1)$$

and the population survival probability $p_{\text{surv}}(\mathbf{n}) = \sum_{k=0}^{K} p_k(\mathbf{n})$.

On the event that the fittest surviving individuals carry $k$ mutations, the asymptotic proportion of $k + i$-mutants in the population is $e^{-u/s_d}(u/s_d)^i/i!$. The asymptotic mean population fitness is thus a random variable $\bar{w}_\infty$ equal to $w_k e^{-u}$ with probability $p_k(\mathbf{n})$ for $0 \leqslant k \leqslant K$, and to 0 otherwise. Therefore

$$\begin{cases} \mathbb{E}(\bar{w}_\infty) = e^{-u} \sum_{k=0}^{K} w_k p_k(\mathbf{n}), \\ \mathbb{E}(\bar{w}_\infty \mid \text{survival}) = e^{-u} \sum_{k=0}^{K} w_k p_k(\mathbf{n})/p_{\text{surv}}(\mathbf{n}). \end{cases} \qquad (2)$$

When going through a bottleneck, this population gives rise to a new founding population with composition $\mathbf{m}$, according to transition probability

$$P(\mathbf{n} \to \mathbf{m}) = \sum_{k=0}^{K} Q_k(\mathbf{m}) p_k(\mathbf{n}) + \mathbf{1}_{\mathbf{m}=\mathbf{0}}(1 - p_{\text{surv}}(\mathbf{n})), \qquad (3)$$

where $Q_k(\mathbf{m})$ is the probability of getting $\mathbf{m}$ supercritical mutants in the sample of size $B$. The resulting transition matrix $P$ then provides an explicit expression of the probability for a viral population with initial state $\mathbf{n}$ to become extinct after going through at most $b$ bottlenecks:

$$p_{\text{bottlenecks},b}(\mathbf{n}) = \mathbf{P}^b(\mathbf{n} \to \mathbf{0}). \qquad (4)$$

In our SEIR epidemiological model, the time spent in the exposed state depends on $\mathbf{n}$ and on the number of Muller's ratchet clicks during the expansion, or equivalently on the type $k$ of the fittest surviving individuals. We show that this incubation period $\tau_{\mathbf{n},k}$ is

$$\tau_{\mathbf{n},k} = \left\lceil \frac{\ln(C/W_{\mathbf{n},k}) - u/s_d}{\ln w_k - u} \right\rceil, \qquad (5)$$

where $W_{\mathbf{n},k}$ is a positive random variable and $\lceil \cdot \rceil$ stands for the ceiling function. As exposed in S1 Text, $W_{\mathbf{n},k}$ can be expressed as the sum of a random number of independent exponential

random variables with common parameter

$$1 + \frac{1}{w_k e^{-u}} W\left[-w_k e^{-u} e^{-w_k e^{-u}}\right],\tag{6}$$

where $W$ stands for the principal solution of the Lambert function.

Simulations were carried out using an agent-based approach in Python. We have created a repository at github.com/strevol-mpi-mis/EvoEpi containing the code and minimum documentation.

Parameter values used in all the simulations reported in this study were chosen to be representative of RNA virus. Genomic mutation rate values, $u$, were taken in the range $0.05 - 0.4$, as described in [31]. [31] compared the DMFE for five different viruses (three with RNA and two with DNA genomes) and found that the mean $s_d$ value was remarkably constant (around 0.11) but with a large variance and a bimodal shape characterized by many mutations of small effect and $20 - 40\%$ of lethal mutations. To match with these values, in our simulations we took $s_d$ in the range $0.05 - 0.9$. Finally, the transmission bottleneck size, $B$, varies greatly across virus and routes of transmission [44], though a consensus exists that very few particles shall be enough to start a new infection [41]. Therefore, we enhance the strength of bottlenecks, we chose $B$ vales in the low range $1 - 10$.

## Supporting information

**S1 Text. Mathematical appendices: Muller's ratchet in expanding populations (A); Survival through multiple bottlenecks (B); Epidemiology of genetic fragility (C); Explicit formulas (D).**
(PDF)

## Acknowledgments

We thank the members of the Structure of Evolution group at MPI MiS for useful discussions.

## Author Contributions

**Conceptualization:** Philip J. Gerrish, Santiago F. Elena, Matteo Smerlak.

**Formal analysis:** Sophie Pénisson.

**Funding acquisition:** Matteo Smerlak.

**Investigation:** Nono S. C. Merleau, Sophie Pénisson, Philip J. Gerrish, Matteo Smerlak.

**Methodology:** Sophie Pénisson, Matteo Smerlak.

**Software:** Nono S. C. Merleau.

**Supervision:** Matteo Smerlak.

**Validation:** Sophie Pénisson, Matteo Smerlak.

**Visualization:** Nono S. C. Merleau.

**Writing – original draft:** Sophie Pénisson, Philip J. Gerrish, Matteo Smerlak.

**Writing – review & editing:** Nono S. C. Merleau, Sophie Pénisson, Philip J. Gerrish, Santiago F. Elena, Matteo Smerlak.

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
