## [Decision Letter · Decision Letter 0]

23 Mar 2021

Dear Dr. Smerlak,

Thank you very much for submitting your manuscript "Why are viral genomes so fragile? The bottleneck hypothesis" for consideration at PLOS Computational Biology. As with all papers reviewed by the journal, your manuscript was reviewed by members of the editorial board and by several independent reviewers. The reviewers appreciated the attention to an important topic. Based on the reviews, we are likely to accept this manuscript for publication, providing that you modify the manuscript according to the review recommendations.

(On a related note, I really enjoyed reading this manuscript myself.)

Sincerely,

Katia Koelle

Guest Editor

PLOS Computational Biology

Sushmita Roy

Deputy Editor

PLOS Computational Biology

[LINK]

Reviewer's Responses to Questions

**Comments to the Authors:**

Reviewer #1: In this manuscript, Merleau and colleagues propose a provocative hypothesis that connects two phenomena commonly observed in virus evolution, namely mutational fragility, and stringent transmission bottlenecks. They propose bottlenecks select for fragile genomes and support this hypothesis through analytical derivation and analysis of a multitype branching process. The authors show that transmission bottlenecks increase the frequency of Muller's ratchet and that mutational fragility protects against extinction in the long term. These results are also put in epidemiological context in an SEIR model in which distinct mutational robustness phenotypes are shown to be better suited to different epidemiological parameters.

The authors provide an elegant theoretical argument for the evolutionary impact repeated bottleneck may have on virus genome design. This is an interesting hypothesis that seems worthy of further exploration. None concerns listed below significantly detract from this achievement. Unfortunately, I am not able to critique the derivations in the supplementary text, although they appear valid and supported by previous work. I will focus my review on the model specification and interpretation.

Figure 1, and the model specification (only deleterious mutations, high mutation rate, no back mutations) seem to suggest that fragile viruses will be selected for in the long run. At least, so long as w_0e^{-u}>1. Bottlenecks only act to speed up the process. Is there a parameter space where bottlenecks are required for fragility to be advantageous? Figure 4 hints at this idea and shows for B=5, w0-1.2, and u=0.05 The survival probability after 1 bottleneck is fairly flat. Is there a parameter space where fragility is selected against, such that some mechanism of mutational robustness is required?

A range of parameter values is used throughout the figures. The manuscript would be strengthened if the parameter values were discussed and could be connected to those observed in viral populations. It appears that relationships between the parameters are more important than the actual values. For example, survival probability depends on whether or not we^{-u}>1, but this could still be put into biological context.

Related to the above point, the SEIR modeling shows how under certain epidemiological conditions fragility can out-compete robustness. However, it seems unrealistic for this selection to take place over 1 epidemic, and this impracticality detracts from an otherwise clear example of the "composite fitness" mentioned in the discussion. Some discussion comparing the parameters in these simulations to those seen in real pathogens would either strengthen the argument that viruses are under the same selective pressures as seen in this model or help to emphasize the purpose of the simulations.

Minor concerns:

1) Figure 1 and throughout the text, the fragile individuals are described as having a smaller starting fitness. It seems the starting fitness of both wild types should be the same (w0).

More time could be spent discussing how fragility evolves. It's not clear that bottlenecks need to be the only or even the main force driving fragility. Shorter genomes with overlapping ORFs likely replicate faster and are fragile. Perhaps, multiple selective forces are pushing in the same direction.

Figure 5) What do the red lines signify?

Methods: After Eq 1 the subscripts in the the series (0,0,0,f_0,f_1,...) are unclear. Should it read (0,0,0,f_k,f_{k+1}), or does i represent the number of mutations more than k?

The SEIR is a nice demonstration of how fragility could be advantageous. It would be a neat (but admittedly unneeded addition) if a parameter space that gave periodic epidemics was used in which this competition could take place over a longer period.

Reviewer #2: This paper proposes a potential explanation for the fragile nature of RNA virus genomes based on population bottlenecks. The authors develop a multi-branching process model, which they use to test the long term viability of fragile vs robust genomes in a regime where several population bottlenecks happen. Moreover, they apply their theoretical formulation to an agent-based compartmental epidemiological model, which confirms the evolutionary advantage of genetic fragility. I find especially the latter part to be very insightful, and potentially of high impact in the emerging evolutionary epidemiology literature. I support publication of the paper, provided the authors address a few minor specific points, highlighted below.

1. I find some similarities between this paper (specifically, the in host evolution) and the analysis of the paper of Stern et al (Stern, A., Bianco, S., Yeh, M. Te, Wright, C., Butcher, K., Tang, C., … Andino, R. (2014). Costs and Benefits of Mutational Robustness in RNA Viruses. Cell Reports, 8(4), 1–11. https://doi.org/10.1016/j.celrep.2014.07.011). An important conclusion of that paper, which is limited to the short term dynamic within a few bottleneck events, is that population fragility gives transient advantage in very selective environments. I wonder whether the author do see this effect in their simulations. I would like the authors to comment on similarities and differences between the two studies.

2. I appreciate the multi-type branching process used to generate the populations. However, it is unclear to me whether or not bottlenecks may create environmental "discontinuity", whereas a population may be subject to stronger or weaker pressure (in the way defined by the authors) following a bottleneck event. I would ask the authors to clarify this point and add it (without to much detail, if they like) to the main manuscript. I apologize if I have not found it.

3. Following my previous comment, since the authors do not define robustness and fragility as property of the environment, I find the epidemiological dynamic unclear. Specifically, if a population transmit from one host to the next, the susceptible host receives n viral particles. My reading of the supplementary information is that there is no additional environmental constraint on the evolving population. However, this host homogeneity is far from being realistic. I would like the authors to comment on this in the manuscript.

**Have all data underlying the figures and results presented in the manuscript been provided?**

Reviewer #1: Yes

Reviewer #2: Yes

PLOS authors have the option to publish the peer review history of their article (what does this mean?). If published, this will include your full peer review and any attached files.

Reviewer #1: No

Reviewer #2: **Yes: **Simone Bianco

Figure Files:

Data Requirements:

Reproducibility:

References:

---

## [Decision Letter · Decision Letter 1]

28 May 2021

Dear Dr. Smerlak,

We are pleased to inform you that your manuscript 'Why are viral genomes so fragile? The bottleneck hypothesis' has been provisionally accepted for publication in PLOS Computational Biology.

Best regards,

Katia Koelle

Guest Editor

PLOS Computational Biology

Sushmita Roy

Deputy Editor

PLOS Computational Biology

Reviewer's Responses to Questions

**Comments to the Authors:**

Reviewer #1: I thank the authors for taking the time to address my concerns and questions. I have no further concerns at this time. Thank you for such an intriguing and thought provoking manuscript.

Reviewer #2: I thoroughly enjoyed reading the paper. I look forward to seeing it out.

**Have the authors made all data and (if applicable) computational code underlying the findings in their manuscript fully available?**

Reviewer #1: Yes

Reviewer #2: Yes

PLOS authors have the option to publish the peer review history of their article (what does this mean?). If published, this will include your full peer review and any attached files.

Reviewer #1: No

Reviewer #2: No

---

## [Editor Report · Acceptance letter]

14 Jun 2021

PCOMPBIOL-D-21-00154R1 

Why are viral genomes so fragile? The bottleneck hypothesis

Dear Dr Smerlak,

I am pleased to inform you that your manuscript has been formally accepted for publication in PLOS Computational Biology. Your manuscript is now with our production department and you will be notified of the publication date in due course.

With kind regards,

Olena Szabo
